# Effects of bear endozoochory on germination and dispersal of huckleberry in the Canadian Rocky Mountains

**Aza Fynley Kuijt**[1]*, **Cole Burton**[1], **Clayton T. Lamb**[2]*

1 Faculty of Forestry, University of British Columbia, Vancouver, BC, Canada, 2 Department of Biology, University of British Columbia, Kelowna, BC, Canada

* clayton.lamb@ubc.ca (CTL); fynley.kuijt@gmail.com (AFK)

## Abstract

Berries are a staple of bear diets during late summer and fall in the southern Rocky Mountains, enabling bears to build up fat reserves and prepare to enter torpor during winter. In turn, bears can benefit fruiting shrubs through dispersal of their seeds. Bears are highly mobile species and seed passage through their guts (endozoochory) can influence seed germination in three ways: deinhibition (removal of germination inhibiting compounds), scarification (mechanical or chemical alteration) and fertilization (enhancement of germination from increased nutrients). We conducted a germination experiment to assess the ways each mechanism of bear endozoochory affects germination success of huckleberry (*Vaccinium membranaceum*.) in the southern Canadian Rocky Mountains. The potential for bears to act as long-distance seed dispersers was also investigated, using a combination of available literature on bear gut retention times and movement data of 74 GPS radio-collared grizzly bears. Deinhibition had a positive significant impact (28.5% germination for the Seeds from Berry treatment compared to 0.2% for Whole Berry at 60 days), while scarification and fertilization did not have detectable positive effects on huckleberry germination success. These results suggest that the removing germination-inhibiting compounds in berry pulp is the primary mechanism through which endozoochory can increase germination in huckleberry seed. We estimated that 50% of the seeds defecated by bears in the region are dispersed 1.1 km away from feeding places (and up to 7 km). The surfaces covered by the seed shadow was up to 149.6 km$^2$, demonstrating that bears can act as effective vectors of seeds over long distances. Endozoochory bolsters the germination success of seeds from fruiting shrubs, and enables seeds to spread to new locations using bears as dispersal agents. Development, resource extraction, and climate change may disrupt the beneficial relationship between bears and huckleberries, where huckleberries help bears gain fat, and bears help spread huckleberry seeds—a process that may become increasingly important as climate change alters habitats.

**Data Availability Statement:** All data and code used for the analysis is available at https://github.com/ctlamb/Huckleberry_Germination_Dispersal_Experiment.

**Funding:** The author(s) received no specific funding for this work.

**Competing interests:** The authors have declared that no competing interests exist.

## Introduction

Frugivory, the consumption of fruits by animals, plays a crucial role in ecosystem functioning by facilitating seed dispersal. Seed dispersal is a vital process that influences plant population dynamics, community structure, and genetic flow [1, 2]. Through seed dispersal, frugivores contribute to the regeneration and expansion of plant populations by moving seeds away from the parent plant, thereby reducing competition and promoting colonization of new areas [3–5]. This mutualistic interaction benefits both the animals, which gain nutrition from the fruits, and the plants, which achieve greater reproductive success.

The effectiveness of seed dispersal is not solely determined by the movement of seeds; it also depends on post-dispersal processes such as seed germination and seedling establishment [3, 6]. The process of frugivory can affect the qualitative aspects of seed dispersal, particularly germination, as the passage of seeds through an animal's digestive system can influence their viability [7]. Assessing the germination success of seeds after ingestion is therefore critical to understanding the full impact of frugivores on plant populations.

Traditional seed dispersal studies often overlook several factors that can affect dispersal outcomes, such as long-distance dispersal events and the unique contributions of large frugivores. Large animals like bears are particularly important because they can move seeds over long distances, providing opportunities for plants to colonize new and potentially more suitable habitats, especially in the context of climate change [3–5]. Despite their significance, the role of large frugivores in seed dispersal remains underappreciated, especially in temperate ecosystems, warranting further investigation.

In this study, we focus on the mutualistic relationship between bears and huckleberries (*Vaccinium membranaceum*) in the southern Canadian Rocky Mountains. Huckleberries are fire-adapted, found in open-canopy forested areas with acidic soils and regularly consumed by numerous organisms including humans and bears [8–10]. Huckleberries are one of the main berries that ripen later in the summer in the southern Canadian Rocky Mountains. As a result, huckleberries represent a substantial portion of the food source available to bears in these areas during late summer [10, 11], making it a genus whose reproductive and dispersal ability may be heavily influenced by consumers. In addition to directly moving fruit seeds, consumers can influence fruits as they pass though the gut, a process known as endozoochory.

Across the northern hemisphere, vast quantities of fleshy fruits are consumed, defecated and dispersed by bears annually. Berries are an invaluable part of the diets for both grizzly (*Ursus arctos horribilis)* and black bears (*Ursus americanus*) and can represent up to 90% of their diet during the summer months [11–13] Berries are an especially important source of carbohydrates in late summer and fall while bears are attempting to gain sufficient fat reserves that will support them [14], and possibly cubs born in the den [15–17]. Berries are often consumed whole by bears, with limited maceration. The small seeds contained in the berries are thus often passed through the gut intact and remain viable [16, 18, 19]. While not strictly necessary for the germination of huckleberry, bear endozoochory has the potential to alter the germination success of seeds, as well as increasing seed dispersal [6, 16, 20–22].

The effects of digestion on germination differs according to the animal and plant species involved. Previous studies on huckleberry species reported that endozoochory enhanced germination [7, 16, 21]. However, the main focus of most of this past research was on the effects of scarification, the mechanical or chemical alteration of the seed coat during digestion. While scarification can influence the germination success of seeds, it is only one of several possible effects affecting germination that could be considered. By consuming berries from fleshy-fruiting shrubs, frugivores such as bears can affect seed germination in three ways: i) alteration of the seed coat due to chemical or mechanical processes during digestion (scarification effect),

ii) removal or elimination of germination inhibiting compounds associated with the pulp of a berry (deinhibition effect) and iii) enhancement of germination with nutrients available from faecal material (fertilization effect) [23–27].

Dispersal of seeds in berries can also be beneficial to a shrub species, providing an opportunity for seeds to germinate away from their parent plants, reducing the chance of self

-fertilization, and increasing gene flow between populations [3, 28]. When deposited by bears in new locations, seeds also have a chance to spread to novel areas with conditions that may be more favourable for germination and growth. In addition, climate change is predicted to alter the suitable habitat of huckleberry, creating a situation whereby seed dispersal will be needed to support establishment of emerging unoccupied habitats [29].

The gut retention time and travel patterns of bears directly relate to their capacity to act as a dispersal agent for fruiting shrub species [5, 30, 31]. Seed shadows, the area of potential dispersal for a seed consumed by a frugivore, can be calculated using gut retention time and frugivore travel data to describe dispersal capacity [30]. Species that retain seeds in their digestive tracts for longer periods of time and travel over large areas have relatively greater capacity to act as long-distance seeds dispersers. Larger species, such as bears, that exhibit long gut retention times and large travel distances could have greater importance in long distance seed dispersal than smaller species with more local ranges and shorter gut retention times [5, 7, 32].

Following previously described methods [16, 24, 25] and building on these works by incorporating improvements to the whole berry treatment (suggested by Robertson et al. [24] and Samuels & Levey [25]), our experiment aimed to isolate the deinhibition, scarification and fertilization effects as mechanisms of change in germination success patterns (Fig 2) and compare results to previous studies and increase the role of replication in science. Previous studies conducted on other species have found scarification to have less of an influence on germination success than deinhibition [16, 21, 24, 33–35]. Based on the results shown by these previous studies, we predicted that the deinhibition effect, where germinating inhibitors within the pulp are removed during endozochory, would have the greatest effect on germination. Germination was predicted to benefit from fertilization by scat, while the scarification effect, where passage through the gut removes protective seed coatings that inhibit germination, was predicted to influence germination the least. We combine the germination results with inference from a rich dataset of collared grizzly bears in the area and previously published gut retention times [31] to assess the ecological relevance and spatial extent of seed dispersal by bears.

## Materials and methods

### Study area

The experiment was performed in the Canadian Rocky Mountains (Fig 1), with samples for the germination trial being collected in Waterton Lakes National Park, Alberta (WLNP) and the Elk Valley region of British Columbia, from July 2021 to September 2021. Mountains surrounding the Elk Valley and mountains within WLNP exhibit habitat suitable to huckleberry, with open-forested landscapes created through a mix of montane forest type, logging and wildfire. Past wildfires have burned much of the area in WLNP where berry and scat samples were collected. The Kenow fire burned over 38% of the park in 2017 [36]. In the Elk Valley, logging has altered many slopes over the last 5–10 years. In the summer of 2021, according to observations by local community members, huckleberry shrubs in both WLNP and the Elk Valley experienced an average to an above-average year for berry production, especially in shaded areas. Shrubs in open areas were more exposed to the extreme heat and record-breaking temperatures experienced during the summer of 2021, which likely contributed to the difference in berry abundance.

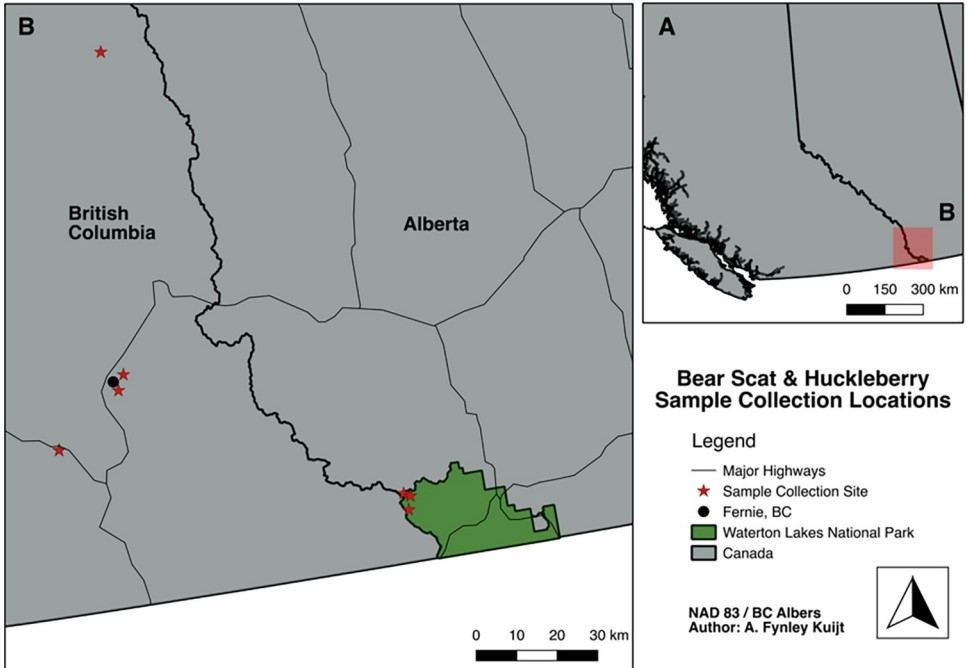

**Fig 1.** Overview map (A) shows the location of East Kootenay, BC and Waterton Lakes National Park, AB within the larger landscape of Canada. Inset map (B) shows bear scat and ripe whole huckleberry sample collection sites within the Elk Valley, BC and Waterton Lakes National Park, AB. Samples were collected opportunistically during July and September, 2021.

## Methods

We opportunistically collected grizzly and/or black bear scats while travelling on hiking trails, off-trail and around Waterton, AB in WLNP and Fernie, BC in the Elk Valley (Fig 1, Table A1 in S1 Appendix). A total of 6 scats were collected that contained huckleberry seeds. An effort was made to find additional scat samples in order to increase the sample size, however bear scats containing huckleberry seeds proved challenging to locate. Scat samples were rated using a three-point scale to estimate scat freshness. Ripe whole huckleberries (n = 208 berries) were also collected from a variety of bushes nearby collected scat samples. Scat and berry samples were placed in paper bags and stored in a refrigerator at 4°C to achieve a cold stratification of approximately 90 days.

The germination trial consisted of four treatments which including planting: 1) Whole Berry: a single, whole berry picked from a shrub, 2) Seeds from Berry: seeds extracted from a berry picked from a shrub, 3) Seeds from Scat: seeds extracted from scat, 4) Mixed Scat: seeds from scat mixed with some scat (Fig 2). Comparing germination rates between these treatments allowed us to differentiate the effects of deinhibition, scarification and fertilization. Comparing the Whole Berry and Seeds from Berry treatment groups was designed to show the influence of the removal of berry pulp and any of its germination inhibiting compounds (deinhibition effect) on germination success. The difference between the Seeds from Berry and Seeds from Scat treatments shows the impact of mechanical or chemical scarification during digestion (scarification effect). Finally, comparing the Seeds from Scat and Mixed Scat treatments displays any enhancement of germination success due to fertilization from the scat (fertilization effect).

The germination trial occurred over two months, from November 29th, 2021 until January 29th, 2022. Past studies looking at huckleberry germination have monitored germination

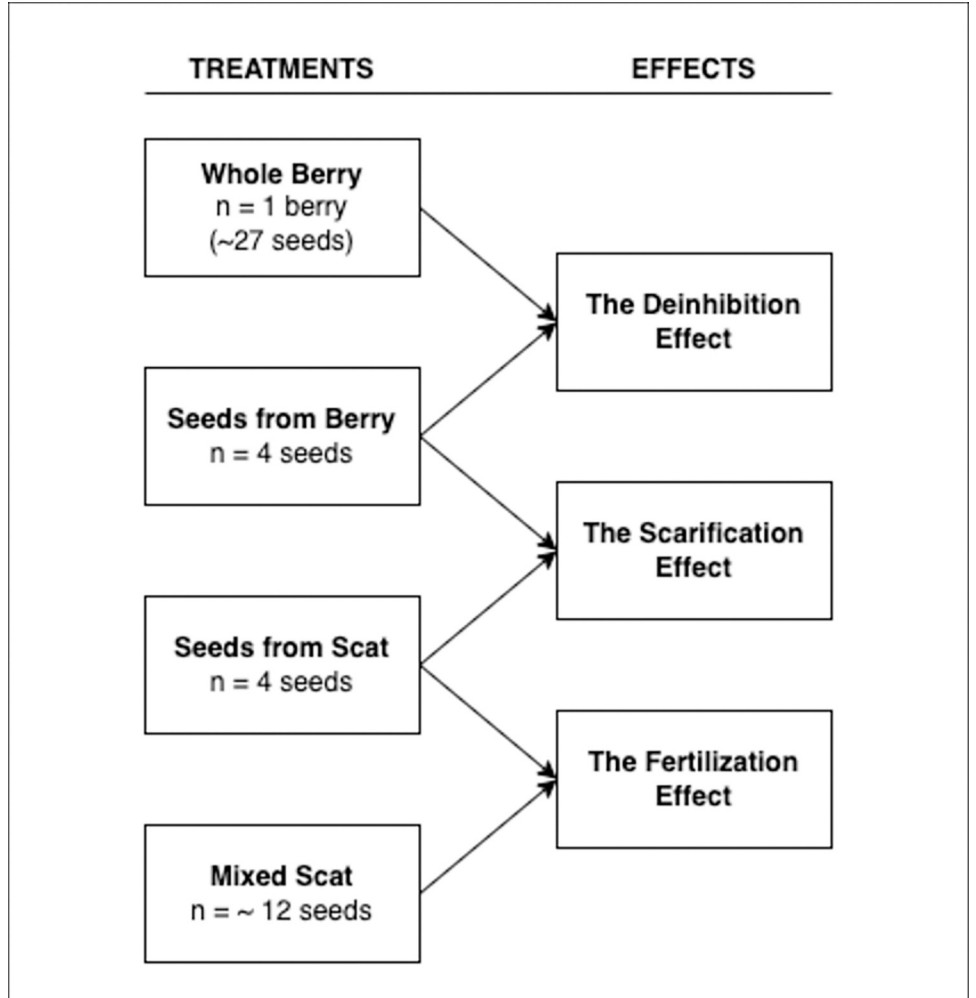

**Fig 2. The four treatments of the germination trial (Whole Berry; Seeds from Berry; Seeds from Scat; Mixed Scat), and their corresponding effects (the deinhibition effect; the scarification effect; the fertilization effect).** Each treatment was replicated 108 times, for a total of 432 samples.

during a four week period [16]. We extended our trial to eight weeks which allowed us to assess the effect of time on observed differences in germination rates between treatments. We conducted the trial indoors in a temperature and light controlled environment in Fernie, British Columbia. Half of the whole berries collected were set aside for planting as the Whole Berry treatment group. Because the number of seeds can be variable for some *Vaccinium* species [37], we determined the number of seeds in a huckleberry by cutting open five whole huckleberries and counting their seeds (23, 31, 26, 38, 21 seeds per berry). On average, the number of seeds per whole berry (27.8) was comparable to the average of 26.67 seeds/berry reported by Nowak and Crone [16]. Huckleberry seeds were manually extracted using tweezers from the other half of collected whole berry samples to create the Seeds from Berry treatment. Seeds were also manually extracted from scat samples for the Seeds from Scat and Mixed Scat treatments. All remaining pulp was removed from the extracted seeds.

We planted all treatments in rehydrated, peat-based Jiffy pellets (Green Garden Products, Washington, USA) under a small amount of growing medium in the germination trial. Jiffy pellets offer a consistent and controlled environment for germination trials, minimizing

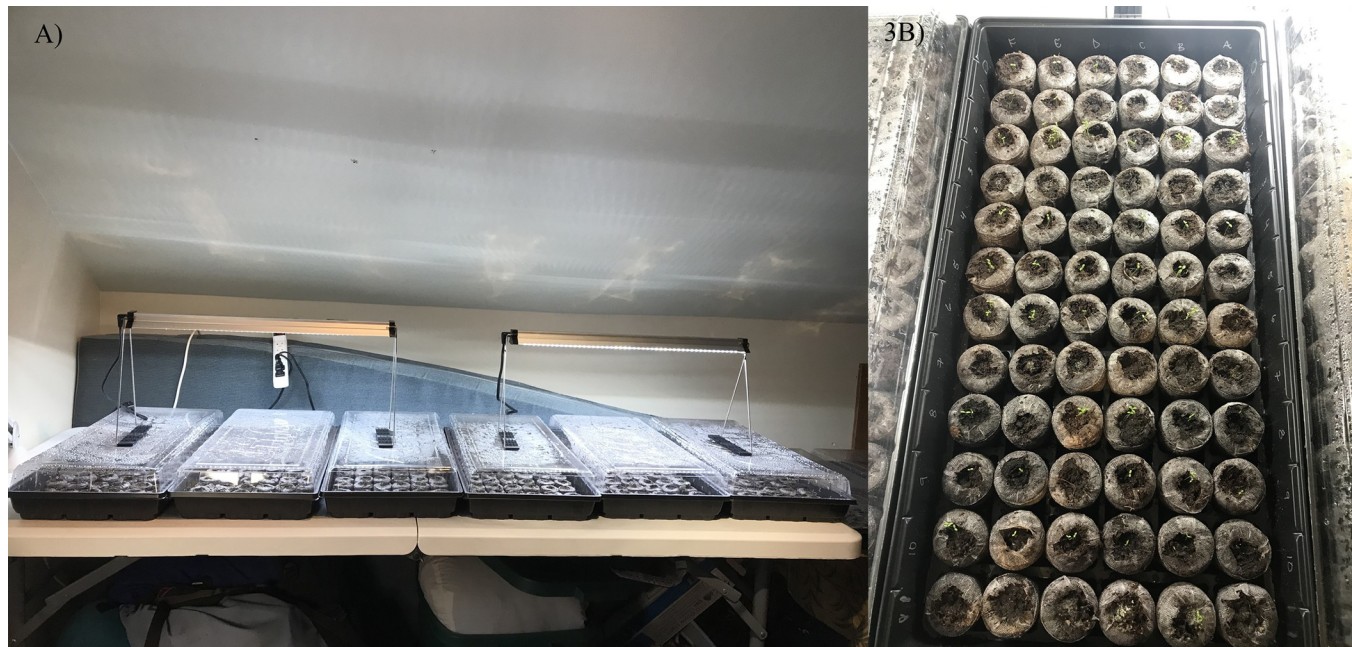

**Fig 3. A.** Indoor experimental setup for the huckleberry germination trial. Six Jiffy germination trays were used, for a total of 432 pellets. Trays were rotated weekly under growing lights for uniform lighting and temperature was controlled to 23°C. **B:** Jiffy germination tray experimental setup. Seeds were planted in each jiffy pellet (see Fig 2 for number of seeds by treatment), in rows randomized by treatment.

variability in soil composition, moisture retention, and nutrient availability. This is crucial for ensuring that observed differences in germination rates are due to the seeds themselves and not variations in the substrate. While the natural substrate that huckleberries would typically encounter in the forests and subalpine of southeast British Columbia is more complex and consists of well-drained soils, rich in organic matter, peat substrates do offer some similarities, particularly in terms of organic content and moisture retention.

The trial consisted of 108 pellets per treatment, sorted among 6 trays of 72 Jiffy pellets each (a total of 432 pellets) (Fig 3A and 3B). The 2 cm radius Jiffy pellets typically rehydrated to ~5 cm high, or a volume of ~75 cm$^2$. The substrate itself. To ensure a random distribution of treatments among each tray, we randomly assigned treatments to the first row of a tray, and then rotated through treatments, row by row, to fill the tray (Fig 3B). Four seeds were planted in each Jiffy pellet for the Seeds from Berry and Seeds from Scat treatment. For the Whole Berry treatment, one berry was planted per pellet. An estimate of the average number of seeds per portion of scat was calculated to be 12 seeds per scat portion, since it was impossible to remove all seeds from the portion of scat added to the Mixed Scat treatment. Thus, approximately twelve seeds were planted in each Jiffy pellet for the Mixed Scat treatment along with a berry-sized pinch of scat planted in addition to the seeds.

## Germination trial

The Whole Berry treatment had approximately 2880 seeds in the germination trial (mean # seeds/berry (26.67) x 108 pellets). The Seeds from Berry and Seeds from Scat treatments were estimated to have 432 seeds each (4 seeds per pellet x 108 pellets). The Mixed Scat treatment was estimated to have 1296 seeds (12 seeds per pellet x 108 pellet).

During the germination trial, growing lights were placed above each planting tray to provide uniform light for 12h a day (Fig 3A) with a constant room temperature of 23°C. We chose

temperature and light conditions to mimic growing conditions near the study area during the summer growing season. Trays were watered as needed to ensure Jiffy pellets were moist to touch. Trays were assessed daily for germination, with germination defined as the first date a plant became visible above the soil. Trays were weekly spun in place and rotated as if in circle, across the table from end to end. We also took note of abnormalities each day such as mold development.

## Seed dispersal

We investigated the spatial extent over which grizzly bears could disperse seeds using published gut retention times (GRT) and movement data from 76 GPS-collared grizzly bears in the Elk Valley of British Columbia [38]. Bear captures were in accordance with the University of Alberta Animal Ethics Committee #AUP00002181 and Province of British Columbia Capture Permit #CB17-264200. Our analysis focused on bear-years during which individual bears were predicted to be in huckleberry-rich areas (areas with >800 huckleberry calories per 900 m$^2$) in August, based on Lamb [39]. The data was filtered using R [40] to identify these specific periods.

To assess the potential seed dispersal distance, we selected locations where bears began in huckleberry patches with >1000 calories per 900 m$^2$ and calculated the distance moved 5–7 hours and 12–18 hours after the initial location. The 5–7 hour period reflects the average GRT when bears had deposited 50% of their cumulative fecal weight following an initial feeding of berries (GRT50%), and the 12–18 hour period represents the maximum time for deposition (GRTmax) as per Elfström et al. [31]. We used the interquartile ranges of GRT to account for individual variability [31] and to provide a broader window to align with collar periods due to missed GPS fixes or differing fix rates.

We estimated the total distance moved as the net displacement from the initial location to the subsequent location within the GRT periods. Given that the telemetry data did not perfectly match the median GRT times reported by Elfström et al. [31], we calculated the speed (m/h) for each identified movement and used these speeds to estimate displacement. For each GRT period, we created 1000 bootstrapped datasets of the speed data, calculating the average and maximum speed observed for each bear year. These values were then averaged to obtain a population mean for each period.

Using these mean and average maximum speeds, we calculated the expected movement distances for the median GRT50% (6:15 hours) and GRTmax (15:38 hours) as reported by Elfström et al. [31].

## Data analysis

After collection, we organised germination trial data using Microsoft Excel and used R Studio Ver. 1.2.5033 [40] to calculate the summary statistics of the total proportion of seeds germinated per treatment. A generalized linear mixed model and post-hoc multiple comparison test was used to test the null hypothesis of no difference in the total proportion of seeds germinated between the designated pairs of treatment groups (Fig 2). We input the number of seeds that germinated as the response variable and a weighted the response by the number of seeds initially planted so that a proportion was effectively estimated using a binomial model with logit link. The mixed model included fixed effects of treatment group, days after planting (30 days and 60 days) and a random intercept for each scat or berry sampling site and a random intercept for each Jiffy pellet to account for the repeated measures. Mixed models were fit in R using the 'lme4' and the multiple comparisons test was done using the 'multcomp' package and the 'glht()' function.

## Results

The overall differences in germination rates among treatment groups was the similar at 30 days and 60 days since planting (Fig 4). The Whole Berry treatment had the lowest germination success, with less than one percent of seeds germinating in 30 days, and 2% germinating in 60 days (Table A2 in S1 Appendix for germination trial data). The Seeds from Berry treatment had the highest germination, with 29% of samples germinating within 30 days and 44% within 60 days. Seeds from Scat and Mixed Scat treatments had intermediate germination rates, with 15% and 12% of samples germinating over 30 days and 31% and 27% germinating over 60 days, respectively. Mold became noticeable in some of the treatments, primarily the mixed scat treatments, within two weeks after planting and remained present throughout the germination trial.

The large and statistically significant difference between the Whole Berry and all other groups (Fig 4, Table A3 in S1 Appendix) indicated that, of the three effects, deinhibition likely had the greatest positive impact on germination (estimated difference in log odds = 4.0–5.6, $p < 0.001$). The smaller and non-significant difference observed between the Seeds from Berry and Seeds from Scat treatments likely indicates that the scarification effect did not strongly influence germination (estimated difference in log odds = 0.88, $Z = 1.49$, $p = 0.25$). Seeds from Scat appeared to germinate at marginally higher, but statistically significant, rates than Mixed scat (estimated difference in log odds = 0.73, $Z = 2.97$, $p < 0.01$) suggesting that fertilization effect may be inhibiting the germination success of huckleberry seeds. The Mixed Scat treatment exhibited significantly higher germination success than the Whole Berry treatment (estimated difference in log odds = 4.0, $Z = 6.4$, $p < 0.001$) with these two treatments being the only two that would reliably occur in nature without human intervention.

Germination of Seeds from Berry, Seeds from Scat and Mixed Scat treatment groups began around the same time, 15 days after planting (Fig 5). However, the Whole Berry treatment did not germinate until 30 days after planting. All treatment groups showed an initial surge in germination, which slowed around 40 days after planting and reached an asymptote at 55 days after planting.

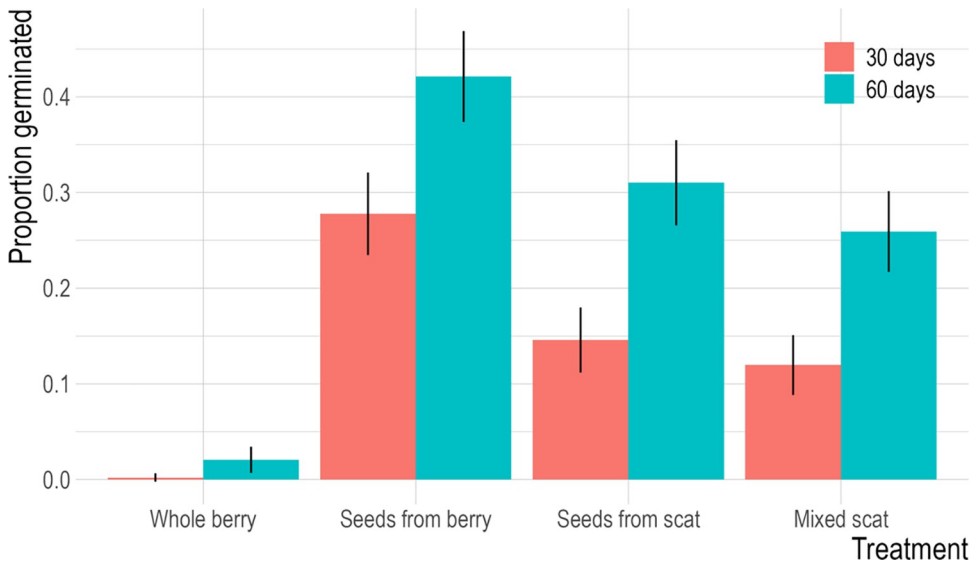

**Fig 4. Total proportion of huckleberry seeds germinated after 30 and 60 days for each of four treatments (Whole Berry; Seeds from Berry; Seeds from Scat; Mixed Scat).** Proportions were calculated using an estimate of total seeds per treatment group. Error bars represent standard error of the proportion.

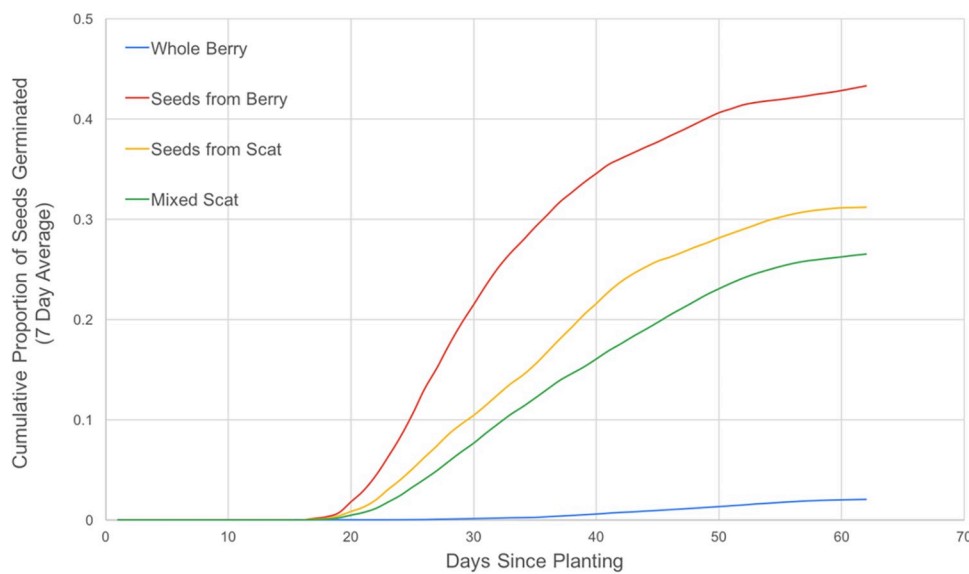

**Fig 5. Cumulative proportion of huckleberry seeds germinated over a 60-day period, for each of the four treatments (Whole Berry; Seeds from Berry; Seeds from Scat; Mixed Scat).** Cumulative proportion of germination was smoothed using a 7-day average.

Summarising the grizzly bear data from Lamb et al. [38], where bears were putatively foraging on huckleberries, resulted in a total of 3,545 bear travel events that met selection criteria and were then used to summarize bear travel for $GRT_{50\%}$, and 3,640 travel events for $GRT_{max}$. Over the 6 hours and 15 minutes ($GRT_{50\%}$) period, bears moved an average of 1.10 km (SE = 0.03) away from potential huckleberry feeding areas in August and September, with an average maximum displacement of 6.92 km (SE = 0.22). Over the 15 hours and 38 minute ($GRT_{max}$) period, bears moved an average of 2.10 km (SE = 0.05) away from potential huckleberry feeding areas in August and September, with an average maximum displacement of 10.49 km (SE = 0.26).

## Discussion

### Seed digestion

This study demonstrates that bear ingestion facilitates huckleberry seed germination, suggesting that scarification is the main process involved. In addition, this study confirms that grizzly bears may disperse a significant proportion of the seeds they consume over distances greater than 1 km away from the feeding areas. The Whole Berry and Seeds from Berry treatments differed significantly, with Seeds from Berry germinating with 387-fold greater success over 60 days, suggesting that the deinhibition effect had the greatest positive impact on germination success. The scarification and fertilizer effects did not have detectable positive effects on germination. Rather, fertilization appeared to have a small negative effect on germination compared to seeds planted without scat. It is possible that fertilization effects would be more pronounced in seedling survival and growth rate rather than germination [41]. The ecological effects of endozoochory were clear: compared to seeds within an intact berry, a seed that has passed through a bear (Seeds from Scat or Mixed Scat) had a more than 80-fold greater chance of germinating.

A small or non-significant scarification effect, compared to the pronounced deinhibition effect is a pattern that has been documented in previous endozoochory studies, conducted on an assortment of fruiting species and frugivores, from birds to bears [16, 21, 24, 33–35]. In this

past research, the nature of the scarification effect has varied from positive to negative to insignificant with different species. Among many factors, seed size is thought to play a dominant role in variation in the effect of scarification as the amount of pulp, thickness of the seed coat, and gut retention times vary between fruits with different seed sizes [33]. Huckleberry seeds are small, especially relative to the amount of pulp around them, consistent with the effects observed in our work where the removal of this pulp (deinhibition) during digestion produced the largest benefits to germination. Indeed, this pattern has been observed in multiple *Vaccinium* species consumed by bears [16, 42].

To separate the interactions of the deinhibition effect, the scarification effect and the fertilization effect, seeds were removed from both huckleberries and scat, to create two treatments (Seeds from Berry & Seeds from Scat). In nature, neither of these artificially created Seeds from Berry or Seeds from Scat scenarios normally exist, since the chance for a seed to be fully separated and planted without intentional human intervention is unlikely. When comparing the two treatments that do occur in nature, germination success with the Mixed Scat treatment was significantly higher than the Whole Berry treatment, providing evidence that endozoochory contributes to increased germination rates for huckleberry in the wild.

It is notable that the pattern of significant differences among all the treatment groups was the same at both 30 days and 60 days after planting. Previous studies on fruiting shrub species, for example by Nowak & Crone [16], have conducted germination trials over only a 30 day, or 4-week period, while others have conducted trials over many months or multiple years [21, 24, 33–35]. Based on the identical relationships of statistical differences among treatment groups throughout the course of the 60-day germination trial in this study, the length of the trial conducted did not substantially alter the results. With cumulative germination levelling off at 55 days, or two months after planting, germination trials of two months appear to be ideal for showing most of the germination that will occur in a short time frame.

## Seed dispersal

From late summer to early fall, the diet of many bears, including those in the Rocky Mountains consists mainly of berries [11–13, 43]. Studies of captive bears fed a diet of berries reported a gut retention time when half of the cumulative fecal weight was deposited ($GRT_{50\%}$) of approximately 6 h and 15 minutes [31]. Applying this gut retention time in tandem with high resolution tracking data from Lamb et al. [38], revealed that grizzly bears in the southern Rocky Mountains could travel an average of 1.1 km from their ingestion sites by the time they had defecated of 50% of their fecal volume, with an average maximum displacement observed across individuals of 6.9 km. With the radius of average bear movement over $GRT_{50\%}$ being 1.1 km, huckleberry seeds have an average dispersal area of 3.8 km$^2$ and an average maximum area of 149.6 km$^2$, also known as the seed shadow. Using the maximal gut retention time, bears could disperse seeds across net displacements of 2.1–10.5 km. These estimations use straight-line displacement to measure the distance travelled by bears. However, bears do not typically travel in a linear fashion and instead wander based on terrain, food availability and many other factors and thus move further across larger areas than these net displacement metrics convey. Though these measurements do not consider the full complexity of bear movements, they provide a basic estimate with which to capture bear movement and seed dispersal capacity across space.

The spatial extent of grizzly bears seed dispersal estimated in our study is consistent with the results from studies across the world. For example, Lalleroni et al., [5] assessed the seed dispersal potential for three brown bears (also *Ursus arctos*) in southwest Europe using the same GRT's as our study. Their results closely mirror ours with the average $GRT_{50\%}$ between 1.09–1.34 km, similar to our estimate of 1.1 km. Both our study and that of Lalleroni et al. [5]

estimated dispersal distances 2–3 fold farther than Koike et al. [30], who assessed Asiatic black bear (*Ursus thibetanus*) movements in Japan. Grizzly bears are known to be fruit consumers across their global range and their ability consumer large volumes of seeds and disperse them multiple kilometers highlights their important role as seed dispersers. Importantly, the relative role of each species in dispersing seeds must consider not only how far or how many seeds are dispersed, but also the density of the disperser on the landscape and the proportion of dispersed seeds that germinate and eventually turn into adult plants [42].

Of the species involved in frugivory in North America, bears have one of the longest gut retention times. Birds, for example, have a $GRT_{max}$ of only 0.3 h to 2.0 h [32], though they also can travel at faster rates. Longer gut retention times correspond with greater potential seed shadows, with the 15h $GRT_{max}$ of bears providing ample opportunity for bears to aid in seed dispersal. The seed shadow is also influenced by a species' movement capability, which in conjunction with gut retention time, demarks the area of potential seed dispersal. Studies have shown that differences in bear gut retention time do not affect seed viability [7]. Bears are also known to defecate several times before passing all the seeds eaten from an area [7], which not only reduces potential competition between seedlings in a deposition area, but also further increases dispersal of seeds within the seed shadow [44].

Bears play an important role as vectors of seeds over large distances in North America as highly effective dispersal agents for fruiting shrub species and by creating complex seed shadows through non-linear movement and periodic defecation [5, 30, 45]. By dispersing seeds, bears also aid in gene transfer between otherwise isolated shrub populations as well as acting as a vector for depositing seeds in new environments. The distribution of huckleberry is predicted to shift substantially under climate change [29]. The dispersal of huckleberry seeds by frugivores will support the species in keeping pace with climate change, which is predicted to shift at ~1–10 km/decade in southeast BC [46]. Grizzly bear dispersal of huckleberry seeds in the Rocky Mountains is on the order of up to 6–11 km per feeding event and will support and likely exceed the minimum movement rate that huckleberry plants will need under climate change. Many other seed dispersers are also likely to support plant dispersal as climates change. For example, González-Varo et al. [47] show that pine marten (*Martes martes*) can disperse fruit seeds an average of 106 m upslopes, exceeding the 35 m/decade warming threshold expected under climate change. Naoe et al. [4] show similar upslope movements of seeds by black bear. There is thus substantial evidence that multiple vertebrates consume, disperse, and support germination of fruit seeds and this process is likely to become even more important in abating deleterious effects of climate change on the distribution and population viability of fruiting species.

## Sources of error

This experiment was conducted in an environment where random variables with the ability to influence germination success, such as temperature or light, were controlled as much as possible. However, the germination trial was not conducted in a traditional germination chamber, lab, or field setting. The unique nature of the germination trial environment may have influenced the results of this study in unknown ways. Regardless, any beneficial or adverse effects would have been evenly experienced across all treatment groups, as care was taken to ensure all trays were rotated regularly and given similar treatment throughout the germination trial.

Mold was present on some Jiffy Pellets within the first two weeks of the trial, with the Mixed Scat treatment having the highest prevalence noted. If mould spores and/or propagules are present in bear scat, bears may also be dispersing mold alongside seeds. It is unknown if the presence of mold altered the germination of seeds. Thus, this is a source of uncertainty in our results, a challenge which others have also faced in similar endozoochory trials [24].

Another source of potential error was the varying number of seeds in each Jiffy pellet. We attempted to controlled for the number of seeds planted in each treatment (Fig 2) by conducting our statistical tests on the proportion of seeds that germinated. We used the average number of seeds present in each Whole Berry and Mixed Scat treatment. It is likely that the actual number of seeds varied between individual Jiffy pellets in these treatments, suggesting some level of unaccounted variation in our results. This level of variation is not expected to change the overall results, especially where the effect sizes are large such as the comparisons with Whole Berry. Similarly, the treatments could have been differentially affected by competition between seeds. There is evidence that germination rates for some *Vaccinium* species can be affected by seed density [3]. It would be wise for future works to match the number of seeds across trails, at least in a portion of the samples, to assess the effects of competition. On the other hand, differing density of seeds is part of the endozoochory process as we have shown here with the approximately 27 seeds within each berry, and only 12 seeds within a similar volume of scat. Thus, while controlling for competition in a portion of trials in future work would be wise, we also urge investigators match real world conditions in a portion of their treatments to allow for estimates of the expected effect size in nature.

In this study, all seeds used in the germination trial were extracted from six different bear scat samples and three whole berry samples. The sample size of scats used in this study was determined by the number of bear scats that were opportunistically found during the months of July to September 2021. Only bear scats that contained huckleberry seeds could be used in this study, which further limited the number of scats collected. Future research should consider using a greater sample size of scats, if available.

## Conclusions

Our results suggest that endozoochory affects the germination of huckleberry in the southern Canadian Rocky Mountains by increasing germination success through deinhibition. Fertilization and scarification did not increase the germination success of huckleberry. Grizzly bears have the potential to be long-distance seed dispersers, spreading seeds across seed shadows of up to $128.7 \text{km}^2$.

Bears are increasingly pressured by urban sprawl and development and the increasing use of natural spaces by humans for recreation and industry in the Rocky Mountains [38, 48]. For example, in 2021, 31 bears were euthanized in the Elk Valley due to injury or for posing a danger to humans in communities within the valley [49]. Considering the ecosystem services bears provide to the germination and dispersal of shrubs such as huckleberry, the mutualistic relationship between bears and huckleberry should be recognized as an important ecosystem function. Land managers, wildlife conservationists and the public alike should consider the importance of these mutualistic relationships between wildlife and vegetation. The disruption of these relationships through the reduced presence of bears in an area could have unforeseen negative impacts on huckleberry and other plant species adapted to rely on endozoochory [50].

## Supporting information

**S1 Appendix.**
(DOCX)

## Acknowledgments

We would like to thank the many friends, family members and Fernie locals who gave their time to help find and collect bear scat, especially those who helped extract seeds and count

germinated plants. Additionally, thank you to the Waterton Lakes National Park staff for their help with navigating research permit applications and providing greenhouse space to conduct the initial trial attempt.

## Author Contributions

**Conceptualization:** Aza Fynley Kuijt, Clayton T. Lamb.

**Data curation:** Aza Fynley Kuijt, Clayton T. Lamb.

**Formal analysis:** Aza Fynley Kuijt, Clayton T. Lamb.

**Investigation:** Aza Fynley Kuijt, Clayton T. Lamb.

**Methodology:** Aza Fynley Kuijt, Clayton T. Lamb.

**Project administration:** Clayton T. Lamb.

**Resources:** Clayton T. Lamb.

**Supervision:** Cole Burton, Clayton T. Lamb.

**Validation:** Aza Fynley Kuijt, Clayton T. Lamb.

**Visualization:** Aza Fynley Kuijt, Clayton T. Lamb.

**Writing – original draft:** Aza Fynley Kuijt.

**Writing – review & editing:** Aza Fynley Kuijt, Cole Burton, Clayton T. Lamb.

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
