## [Decision Letter · Decision Letter 0]

24 Jul 2024

PONE-D-24-27272Effects of Bear Endozoochory on Germination and Dispersal of Huckleberry in the Canadian Rocky MountainsPLOS ONE

Dear Dr. Lamb,

Thank you for submitting your manuscript to PLOS ONE. After careful consideration, we feel that it has merit but does not fully meet PLOS ONE’s publication criteria as it currently stands. Therefore, we invite you to submit a revised version of the manuscript that addresses the points raised during the review process.

In general, the reviewers well received the manuscript but highlighted some important comments that should be addressed before further consideration. I ask you to carefully revise the manuscript according to the reviewers' comments ( see also the attached file).

We look forward to receiving your revised manuscript.

Kind regards,

Francesco Boscutti

Academic Editor

PLOS ONE

Journal Requirements:

4. We note that you have referenced (Lamb, C. T., L. Smit, G. Mowat, B. McLellan, and M. Proctor. 2023. Unsecured attractants, collisions, and high mortality strain coexistence between grizzly bears and people in the Elk Valley, southeast British Columbia (in prep). Conservation Science and Practice.) which has currently not yet been accepted for publication. Please remove this from your References and amend this to state in the body of your manuscript: (ie “Bewick et al. [Unpublished]”) as detailed online in our guide for authors

6. We note that Figure 1 in your submission contain map/satellite images which may be copyrighted. All PLOS content is published under the Creative Commons Attribution License (CC BY 4.0), which means that the manuscript, images, and Supporting Information files will be freely available online, and any third party is permitted to access, download, copy, distribute, and use these materials in any way, even commercially, with proper attribution. For these reasons, we cannot publish previously copyrighted maps or satellite images created using proprietary data, such as Google software (Google Maps, Street View, and Earth). For more information, see our copyright guidelines: http://journals.plos.org/plosone/s/licenses-and-copyright.

Reviewers' comments:

Reviewer's Responses to Questions

**Comments to the Author**

1. Is the manuscript technically sound, and do the data support the conclusions?

Reviewer #1: Partly

Reviewer #2: Yes

2. Has the statistical analysis been performed appropriately and rigorously? 

Reviewer #1: Yes

Reviewer #2: Yes

3. Have the authors made all data underlying the findings in their manuscript fully available?

Reviewer #1: Yes

Reviewer #2: Yes

4. Is the manuscript presented in an intelligible fashion and written in standard English?

Reviewer #1: Yes

Reviewer #2: Yes

5. Review Comments to the Author

Reviewer #1: A detailed review of this article has been uploaded as PDF

Review summary:

This study investigates the role of bears in the dispersal and germination of huckleberry (Vaccinium spp.) seeds in the Canadian Rocky Mountains. The authors performed ex-situ germination experiments under controlled conditions to assess the effects of bear ingestion on huckleberry seed germination, distinguishing four different germination treatments. These treatments allowed them to test the effects of deinhibition, scarification, and fertilization promoted by bear ingestion and defecation on Vaccinium seed germination rates. The authors found that deinhibition had the strongest positive effect on seed germination rates, while the other two processes promoted by bear ingestion and defecation (scarification and fertilization) had neutral to moderately negative effects. Additionally, the authors combined published data on bear gut retention times and bear movement from radio-collared grizzly bears to calculate the potential distances at which grizzly bears can disperse the seeds in the Canadian Rocky Mountains. They estimated that the average distance at which bears disperse the seeds exceeds one kilometer and that the potential area of seed dispersal may be up to 150 km². Based on their results, the authors highlight the importance of the ecosystem services provided by bears in the region.

In general, the manuscript is well-written and easy to follow, and the presentation of results is clear. However, I believe the overall quality may still improve with some substantial modifications. The flow of some parts of the document, especially the introduction and the discussion, can be improved by implementing several changes not only in the wording but also in the overall structure. Additionally, I have some concerns about the assumptions made regarding the germination treatments, which makes me wonder about the actual ecological implications of the results obtained.

Reviewer #2: Comments to the Authors

MS. Ref. No.: PONE-D-24-27272

Title: " Effects of Bear Endozoochory on Germination and Dispersal of Huckleberry in the

Canadian Rocky Mountains” Kuijt et al.

This MS examines important aspects of seed and reproductive biology of huckleberry (Vaccinium sp.) through bear endozoochory in Canadian Rocky Mountains. In particular the authors investigated germination success (total germination percentage within 30 and 60 days in Jeffy pellets under controlled laboratory conditions) of four different seed treatments (whole berry, seeds from berry, seeds from scat and mixed scat) and effectively disentangled de-inhibition effect, scarification effect and fertilization effect. They demonstrated that probability of germination in huckleberry seeds in nature was strongly affected by the removal of inhibitory compounds through

pulp removal (de-inhibition effect), whereas scarification and fertilization did not. Additionally, they also studied contribution of bears in successfully dispersing seeds over long distances by combining literature and movement data.

In my opinion, the MS deserves to be published in PLOS ONE since it covers an interesting topic of scientific value and the results presented may provide new comprehension on it. Also, I believe that the authors have considered some specific concerns raised by recent literature (see Robertson et al. 2006, but also Rogers et al. 2021, which has not been cited in References and could be added and be considered in Discussion), where it has been highlighted the need to include intact fruits in germination test for investigation of frugivory impact on seed dispersal.

Also, some minor points that the authors need to reconsider before publication are the following:

Minor points

Line 31: avoid italics in “spp.”

Lines 75 and 97-98: check and correct the style in “García-Rodríguez et al. 2021a,b” and “García-Rodríguez et al. 2022”

Lines 84, 114 and 121: correct author citation in “Robertson”

Lines 92 and 116: in citation of Robertson et al., correct year in “2006”

Line 313: correct in “scarification”

Line 189-190: give information on the final volume of rehydrated Jeffy pellet. Also, the authors could add here more information why they choose to use peat substrate for germination trial and if this substrate simulate or not natural conditions where berries fall from plants.

Line 206-207: the sentence is correct only for the Seeds from Berry and Seeds from Scat treatments. Add correct information for the other two treatments. Regarding this point: I wonder whether the comparison among Whole Berry and Seeds from Berry should be performed taking in account that in the same volume of each Jeffy pellet the authors tested germination of different numbers of seeds (27 vs. 4 seeds). Maybe it could be useful, if possible, to add an additional test to check if germination in Seeds from Berry Treatment does not change if 4 or 27 seeds were planted in one single Jeffy pellets. The same concern could be valid for Mixed Scat Treatment.

Lines 257-258: have the authors checked at the end of germination trial if the not germinated seeds in Whole Berry treatment were still viable or dead? I believe this information is very important to understand if the seeds are effectively dormant and not germinating because of deinhibitory effects from pulp or they died for lack of oxygen or fungal infection during test.

This aspect was also evidenced in Robertson et al. 2006, which could render more difficult to assess the impacts of endozoochory.

6. PLOS authors have the option to publish the peer review history of their article (what does this mean?). If published, this will include your full peer review and any attached files.

Reviewer #1: **Yes: **ALBERTO GARCIA-RODRIGUEZ

Reviewer #2: No

---

## [Author Response · Author response to Decision Letter 0]

13 Sep 2024

Dear Dr. Boscutti,

We are pleased to submit the revised version of our manuscript titled “Effects of Bear Endozoochory on Germination and Dispersal of Huckleberry in the Canadian Rocky Mountains” (Manuscript ID: PONE-D-24-27272). We have carefully considered the insightful comments and suggestions provided by the reviewers, and we have made substantial revisions to improve the clarity, accuracy, and overall quality of the manuscript.

Summary of Revisions:

 1. Abstract: We have revised the abstract to more clearly state the main goal of the study, which is to isolate the effects of three mechanisms—deinhibition, scarification, and fertilization—on the germination success of huckleberry seeds following ingestion by bears. We have also included a brief description of the experimental treatments, making the connection between the treatments and the processes more explicit.

 2. Introduction: Following Reviewer 1’s suggestion, we have restructured the introduction to first present general ecological concepts such as frugivory and seed dispersal, followed by a discussion of seed dispersal effectiveness and the role of large frugivores. We conclude with a focused description of our study system and the main objectives of the research.

 3. Methods: In response to the reviewers’ feedback, we have added detailed descriptions of the study species (Vaccinium membranaceum) and the rationale for using Jiffy pellets in the germination trial. We have also clarified the calculation of seed numbers per treatment and addressed concerns regarding the variation in seed numbers across different treatments.

 4. Results and Discussion: We have revised the discussion to avoid repetition of the introduction and results. The discussion now more clearly integrates our findings with existing literature, highlighting the unique contributions of this study to our understanding of bear-mediated seed dispersal and germination. We have also added new references, including recent studies that support our findings on the role of bears as long-distance seed dispersers in the context of climate change.

 5. Addressing Specific Reviewer Comments: We have made numerous other revisions throughout the manuscript to address specific points raised by the reviewers, such as clarifying the methodology, improving the presentation of results, and refining the statistical analyses. These changes are detailed in our response to reviewers document, which we have also included with this submission.

We believe that these revisions have significantly strengthened the manuscript, and we are confident that it now provides a more robust and insightful contribution to the field of seed dispersal ecology. We sincerely appreciate the time and effort that you and the reviewers have invested in the evaluation of our work, and we hope that the revised manuscript meets your expectations.

Thank you for considering our revised submission. We look forward to your feedback and hope to have the opportunity to share our work with the readers of PLOS ONE.

Sincerely,

Clayton Lamb, PhD

University of British Columbia, Kelowna, Canada

Reviewer's Responses to Questions

Comments to the Author

Reviewer #1: 

ABSTRACT:

My main suggestion for the abstract is to emphasize more the main goal of the study. The authors mention the three processes that may affect seed germination rates after being ingested and defecated by frugivores (i.e., deinhibition, scarification, and fertilization). However, the authors do not clearly specify in the abstract that the study's goal is to isolate the effects of these three processes to describe their individual effects on the fate of the defecated seeds. Additionally, experimental treatments must be briefly described in the abstract. Even if it seems obvious to the authors, the link between the different treatments and the processes should be explicitly described so potential readers can understand why these four treatments were chosen.

- Agreed. We have changed the pitch of the abstract to more concretely link to the goal of the study “We conducted a germination experiment to assess the ways each mechanism of bear endozoochory affects germination success of huckleberry (Vaccinium spp.) in the southern Canadian Rocky Mountains.”

L27: Replace “influence” with “benefit.”

- done

L39: Modify to “can increase germination in huckleberry seeds.”’

- done

L39-42: This sentence needs rewriting. It is more precise to say that around 50% of the seeds defecated by bears in the region are dispersed more than 1 km away from feeding places (and up to 7 km) and that the seed dispersal area can cover up to 150 km².

- done

- 

L41: The definition of GRT50 is imprecise. The authors refer to the time when approximately half of the animals would have defecated, but the concept actually denotes the time when 50% of the cumulative weight of fecal remains had been defecated after the ingestion of specific food items (from Elfström et al. 2013).

 -thank you, we have changed this throughout

L42: As the seed shadow does not only include the surface but also the spatial distribution of seeds within this surface (i.e., different seed densities according to the distance to the feeding place), I recommend rephrasing “the resulting seed shadow” to “the surface covered by the seed shadow.”

- done

L43: The effectiveness of the seed dispersal services provided by a disperser agent is traditionally evaluated based on the quantity and quality of the seeds dispersed, not based on the seed shadow covered by its dispersal services. L45-47: The fact that bear-berry interactions can benefit both bears and berry species is independent of increasing levels of human use and climate change. What is relevant as a final message is that these increasing pressures may threaten the ecosystem services provided by bears, with potential consequences not only for Vaccinium species but also for the entire ecosystem.

 - edited to the following “Development, resource extraction, and climate change may disrupt the beneficial relationship between bears and huckleberries, where huckleberries help bears gain fat, and bears help spread huckleberry seeds—a process that may become increasingly important as climate change alters habitats.”

KEW WORDS: Some key words are already included in the title (bear, endozoochory, huckleberry). Please, replace them by others that are not.

- Removed, and replaced with “Ursus arctos, deinhibition, fertilization, vaccinium, scarification, seed dispersal”

INTRODUCTION:

The entire introduction would benefit from some changes in the order in which the main ideas are presented. For instance, instead of starting by discussing the bear-berry relationship (the first two paragraphs in the current format are about Vaccinium species and the importance of berries for bears), the authors must consider starting by introducing general ecological concepts such as frugivory and seed dispersal, highlighting their vital role in ecosystem functioning. From that, the authors may narrow down the focus, concluding with a final paragraph where they describe their study system and the main objectives of this work. A possible order for a more general introduction could look as follows:

Paragraph 1: Frugivory and seed dispersal: concepts and main benefits for both animals and plants.

Paragraph 2: Seed dispersal effectiveness: the need to assess germination as animal ingestion directly affects the qualitative component of seed dispersal services.

Paragraph 3: Potential determinants of seed dispersal effectiveness traditionally neglected in seed dispersal studies: the importance of long-distance dispersal events, benefits for the plants, and the unique role of large frugivores in providing such services.

Paragraph 4: Brief description of the study system (just a couple of sentences about the relevance of bear-berry interactions) and main objectives of the study. Then, I would move the specific paragraphs about Vaccinium species (the first paragraph in the current

version) and the importance of fruits and particularly berries in the bear diet (the second paragraph) to the methods section, where the authors may add a section called “Study Species,” including two subsections: one about the main characteristics of Vaccinium species and another about bears and the importance of fleshy fruits for these species.

Additionally, I personally miss a deeper description of the three processes potentially affecting seed germination during the ingestion and defecation by animals. For instance, do the authors know about any specific compound promoting deinhibition? Is scarification affected by seed size and seed species? Are the authors aware of any published information about the possible role of scat remains as seed fertilizer?

 - We have now considerably reworked the introduction and added the suggested paragraphs.

L75-77: The last sentence of this paragraph is not accurate. There is a vast amount of literature about this mutualistic interaction. As most of these studies have been conducted either in the tropics or have focused on birds, I recommend the authors to rephrase this sentence to highlight that studies about the role of large frugivores in temperate ecosystems have often been neglected.

- Added “The extent to which endozoochory by large vertebrates provides dispersal and fitness benefits to fruiting shrubs in temperate ecosystems remains a poorly studied mutualistic interaction.”

L78-79: Replace with “differs according to the animal and plant species involved.”

 - done

L83: “it is only one of several…”

 - done

L84-87: The text flow will improve if the authors delete the part about the whole berry treatment. Doing so will directly link the part about the three processes by which endozoochory may affect germination with the previous sentence stating that scarification is only one of these processes.

 - done

L93-94: Please remove the part about the seed treatments from the introduction and explain them in more detail in the methods.

 - done

L110: Greater importance in what context? I would just say that large frugivores contribute most to the long-distance dispersal events. L109-111: As written, it seems that Lalleroni et al. compared seed dispersal distances by different taxa, which is not true. Please rephrase or add extra references that support this comparison.

 - rephrased and added more citations “Larger species, such as bears, that exhibit long gut retention times and large travel distances could have greater importance in long distance seed dispersal than smaller species with more local ranges and shorter gut retention times (Lalleroni et al., 2017, Traveset and Willson 1997, Fukui 2003).”

L113: “and dispersal distances of huckleberry…”

 - done

MATERIAL AND METHODS:

My major concern about this section is regarding some assumptions made by the authors to calculate the germination rates of the different treatments.

In the whole berry treatment, the authors used 5 Vaccinium fruits to estimate the average number of seeds per berry. Regardless of the small sample size (only 5 fruits), which I consider insufficient to provide a robust estimation of the mean number of seeds per berry, I have two doubts that I think are not properly clarified in the manuscript:

1. Since the authors assigned all Vaccinium species as huckleberry (L53), they do not provide any information about the species collected. Did all the collected fruits (including those used for seed counting and those used for the germination trial) belong to the same Vaccinium species? Which one?

This is important because different Vaccinium species may have different average numbers of seeds per fruit.

- helpful point, yes all were the same species and we were too generic in the past draft. We’ve now added the species name of Vaccinium membranaceum into the manuscript

2. The authors collected huckleberry fruits around each of the bear scats collected. However, they do not provide any information about the original location of the 5 fruits collected, nor about the criteria followed to choose the fruits (the only thing mentioned is that they collected ripe fruits). For instance, seed set per Vaccinium myrtillus fruit is highly variable, potentially affected by environmental variables, and with heavier fruits containing on average larger numbers of seeds1-4. If the authors didnot control for fruit size and other factors potentially affecting seed production, then I would assume a significant variability in seed numbers, which could severely compromise the accuracy of the results

obtained.

- From our 5 fruits we obtained a mean of 27.8 seeds/fruit, which is very similar the results of (Nowak and Crone 2012) who counted the seeds of 10 huckleberry fruits and obtained an average of (x~26.67, SD~10.40). While there is variation between fruits our average of 27.8 appears reasonable and consistent with estimates from the same species collected 100-300 km away from our site.

1-García-Rodríguez A., Albrecht J., Frydryszak D., Parres A., & Selva N. (2024). Interactive effects of elevation and canopy affect

bilberry performance in a temperate coniferous region. Plant Ecology, 225(2), 81-91.

2-Pato, J., & Obeso, J. R. (2012). Growth and reproductive performance in bilberry (Vaccinium myrtillus) along an elevation

gradient. Ecoscience, 19(1), 59-68.

3-Ranwala, S. M., & Naylor, R. E. (2004). Production, survival and germination of bilberry (Vaccinium myrtillus L.) seeds. Botanical

Journal of Scotland, 56(1), 55-63.

4-Selås, V. (2000). Seed production of a masting dwarf shrub, Vaccinium myrtillus, in relation to previous reproduction and

weather. Canadian Journal of Botany, 78(4), 423-429.

Similarly, I have serious concerns about the accuracy of the results of the mixed scat treatment. 

In L198-199, the authors explain that they planted about 12 seeds along with a berry-sized pinch of scat. However, in L404-409, the authors admit that this procedure may have led to differences in the number of seeds per pellet, potentially compromising the results obtained. As they aimed at having very similar (or even identical) seed numbers in all the mixed scat treatment pellets, I would have recommended the authors that, instead of selecting “berry-sized” pinches of different scats, they could simply mix/homogenize a unique bear scat in order to have similar seed densities within different portion of the scat, and then add a standardized subsample from this scat (for instance 0.5 grams) to each pellet.

- This is a good idea and something we will consider trying in the future.

Additionally, since the authors planted different amounts of seeds per pellet in each treatment (~27 seeds per pellet in the whole berry treatment, ~12 in the mixed scat treatment, and 4 in the seeds from berry and seeds from scat treatments), possible effects of competition among seeds may have been masked by the effects of the different treatments. For instance, Vaccinium myrtillus seed germination rates are negatively affected by the number of seeds planted together, suggesting that competition among seeds is stronger with increasing seed densities5. In order to discard competition effects, the authors must have planted similar seed numbers in the 4 treatments considered, instead of planting different seed numbers

in each treatment.

5- García‐Rodríguez, A., Albrecht, J., Farwig, N., Frydryszak, D., Parres, A., Schabo, D. G., & Selva, N. (2022). Functional

complementarity of seed dispersal services provided by birds and mammals in an alpine ecosystem. Journal of Ecology, 110(1),232-247.

- This is a fair point. There are different ways to look at the issue I suppose. We were looking to replicate the real world effects of these different mechanisms and competition may interact and be part of the mechanism perhaps. I suppose we could’ve done all treatments to match the mean of the whole berry (27 seeds) to standardize the competition effect. Had we done that there was little difference between whole berry and mixed scat we would likely be wondering if the result was useful because we artificially increased the densities of seeds above the density we see distributed in scats. I think the best we can do here

---

## [Decision Letter · Decision Letter 1]

26 Sep 2024

Effects of Bear Endozoochory on Germination and Dispersal of Huckleberry in the Canadian Rocky Mountains

PONE-D-24-27272R1

Dear Dr. Lamb,

We’re pleased to inform you that your manuscript has been judged scientifically suitable for publication and will be formally accepted for publication once it meets all outstanding technical requirements.

Kind regards,

Francesco Boscutti

Academic Editor

PLOS ONE

Additional Editor Comments (optional):

Reviewers' comments:

Reviewer's Responses to Questions

**Comments to the Author**

1. If the authors have adequately addressed your comments raised in a previous round of review and you feel that this manuscript is now acceptable for publication, you may indicate that here to bypass the “Comments to the Author” section, enter your conflict of interest statement in the “Confidential to Editor” section, and submit your "Accept" recommendation.

Reviewer #2: All comments have been addressed

2. Is the manuscript technically sound, and do the data support the conclusions?

Reviewer #2: Yes

3. Has the statistical analysis been performed appropriately and rigorously? 

Reviewer #2: Yes

4. Have the authors made all data underlying the findings in their manuscript fully available?

Reviewer #2: Yes

5. Is the manuscript presented in an intelligible fashion and written in standard English?

Reviewer #2: Yes

6. Review Comments to the Author

Reviewer #2: Comments to the Authors

MS. Ref. No.: PONE-D-24-27272R1

Title: " Effects of Bear Endozoochory on Germination and Dispersal of Huckleberry in the

Canadian Rocky Mountains” Kuijt et al.

The authors have addressed all comments and the Manuscript could be now considered acceptable for publication.

Check carefully for small errors:

Line 32: delete dot after Vaccinium membranaceum

Line 50: add italics for Ursus arctos

Lines 85: correct though in through

Line 343: check this sentence: “suggesting that scarification is the main process involved”. I think the authors meant that the deinhibition effect is the main process instead.

Lines 405-417: delete underlined text

Lines 460-461: check citation: the name of author should be in brackets

7. PLOS authors have the option to publish the peer review history of their article (what does this mean?). If published, this will include your full peer review and any attached files.

Reviewer #2: No

---

## [Editor Report · Acceptance letter]

16 Oct 2024

PONE-D-24-27272R1 

PLOS ONE

Dear Dr. Lamb, 

I'm pleased to inform you that your manuscript has been deemed suitable for publication in PLOS ONE. Congratulations! Your manuscript is now being handed over to our production team.

Kind regards, 

on behalf of

Dr. Francesco Boscutti 

Academic Editor

PLOS ONE